

**Migration behavior of benzobicyclon hydrolysate and associated influencing**
**factors in different agricultural soils**
Lang Liu[1], Lei Rao[2], Wenwen Zhou[3], Limei Tang[2], Baotong Li[1,*]
[1] *College of Land Resources and Environment, Jiangxi Agricultural University, Nanchang*
*330045, China*
[2] *College of Agricultural Sciences, Jiangxi Agricultural University, Nanchang 330045, China*
[3] *College of Food Sciences, Jiangxi Agricultural University, Nanchang 330045, China*
* Corresponding author. College of Land Resources and Environment, Jiangxi Agricultural
University, 1225 Zhimin Road, Nanchang 330045, China.
*E-mail address*: libt666@163.com (B. Li)
*Abbreviations:* BH, benzobicyclon hydrolysate; BZB, benzobicyclon



**Abstract**
Benzobicyclon is a triketone pro-herbicide that needs to be hydrolyzed to form an active
compound benzobicyclon hydrolysate (BH). This study aimed to investigate the migration
behavior of BH in different types of agricultural soil and the associated influencing factors.
Soil thin-layer chromatography and column leaching tests were used to study the migration
behavior of BH in these soils. Based on the mobility retention factor ($R_f$ = 0.34–0.90), the
mobility of BH in thin soil layers was ranked in the order Lixisols > Anthrosols > Ferralsols >
Phaeozems. The $R_f$ value of BH was linearly positively correlated with soil sand content and
pH, and negatively correlated with other physical and chemical properties of soil. BH was
difficult to leach in Phaeozems, less difficult to leach in Ferralsols, and easy to leach in
Anthrosols and Lixisols. Increasing the BH dosage and rainfall amount or adding humic acid
and anionic (dodecyl benzene sulfonic acid) or nonionic (Tween-80) surfactant blocked BH
migration in soil columns. In contrast, increasing the leaching solution pH and adding cationic
surfactant (cetyl trimethyl ammonium bromide) promoted BH migration in soil columns.BH
application has a low risk of groundwater pollution in Phaeozems and Ferralsols, but poses a
potential threat to groundwater in Anthrosols and Lixisols.
*Keywords:* Benzobicyclon hydrolysate; Herbicide; Leaching; Migration; Soil thin-layer
chromatography






## 1. Introduction

In rice production, the presence of weeds is a serious problem for rice crop growth and high food yields, because weeds and rice plants coexist in paddy fields and compete for resources such as nutrients, light, and growth space (Fartyal et al. 2018). Methods for controlling weeds in paddy fields mainly include manual and mechanical weeding, cropping systems, cultivation measures, and herbicides. In particular, the application of herbicides is effective, economical, time-saving, and labor-saving, and has become the most important control method in the integrated weed management system of paddy fields (Powles &Yu 2010). Although the widespread application of herbicides has increased crop yields, the spread of herbicide residues in the environment also leads to environmental pollution and threatens human health (Carvalho 2017). The environmental behavior of the herbicide in soil includes migration, adsorption, desorption, degradation, and crop absorption, which are the key processes influencing the final fate of the herbicide in the environment (Liu et al. 2020). Because the amount of herbicides directly acting on crop targets is limited, most of them will enter the paddy soil, herbicide residues migrate and diffuse into the water environment of paddy fields, infiltrating the soil and possibly polluting groundwater (Morrissey et al. 2015). Therefore, studying the migration behavior of herbicide residues in the soil is of great significance for the safety evaluation of herbicides, prevention and control of environmental pollution, and protection of human health.

Herbicide leaching in soil (movement performance) refers to the movement of herbicides vertically downwards along with the soil profile with infiltration water. As leaching is an important process influencing the migration and final fate of herbicides in the soil–water



system, evaluating whether herbicides will enter the groundwater and cause environmental
pollution is critical (Younes &Galalgorchev 2000). Studying the migration behavior of
herbicides in soil from an environmental safety perspective can provide insight into their
potential impact on groundwater quality, which has implications regarding the application of
necessary restrictions on the scope and intensity of herbicide application (Sabale et al. 2015).
Many factors influence the migration behavior of herbicides in the soil, including the properties
of the herbicide and soil, the herbicide application methods and dosages, and environmental
factors (Kumari et al. 2020). Generally, herbicides with higher water solubility are much easier
to migrate in soil (Konstantinou et al. 2001). The soils with distinctly different properties also
affect herbicide migration. For example, the migration capacity of herbicides is proportional to
the soil organic matter content (Muhammad Ashraf et al. 2012). Furthermore, the application
dosage of herbicides influences their leaching and migration behavior in soil. Additionally,
after herbicides are applied to agricultural soil, they can easily migrate to groundwater through
rainfall or irrigation water (Laabs et al. 2000). Surfactants are also important parts of herbicide
preparation (Wang et al. 2017) and can be divided into cationic, anionic, and nonionic
surfactants according to whether they dissociate in water to generate ions (Alwadani &Fatehi
2018). As groundwater is a major source of drinking water and irrigation water, understanding
the migration behavior of herbicides in soil under the influence of various factors is important.

Benzobicyclon (BZB) is a bicyclooctane triketone herbicide developed by SDS Biotech

KK (Japan) in 1992. This herbicide inhibits the activity of $p$-hydroxyphenylpyruvate dioxidase
and affects the synthesis of plastoquinone, thereby influencing the biosynthesis of carotenoids
and causing leaves chlorosis (Van Almsick 2009). Owing to its broad spectrum and high activity,





BZB has been widely used to control annual weeds, sedges, gramineous weeds, and broadleaf
weeds in paddy fields globally (Brabham et al. 2019, McKnight et al. 2018). As a pro-herbicide,
BZB itself is inactive, but is rapidly hydrolyzed in flooded paddy fields to become its active
form with herbicidal activity, benzobicyclon hydrolysate (BH) (Williams &Tjeerdema 2016).
Previous studies have shown that BH has high hydrolytic stability and solubility in water, with
a half-life greater than 1 year (25 °C), while the solubility of BH increases with increasing
solution pH (Williams et al. 2017). Furthermore, different agricultural soils have different BH
adsorption values (Willett et al. 2020). These physical properties might lead to the BH leaching
into groundwater or other water bodies, posing a huge environmental risk. The adsorption and
desorption behavior of BH in soil has been reported (Willett et al. 2020). However, its leaching
and migration in soil and associated influencing factors have not been explored.
Therefore, this study investigated the migration behavior of BH and associated
influencing factors in different agricultural soils using thin-layer chromatography and column
leaching tests. This study aimed to analyze the migration characteristics of BH in four different
types of soil, quantify the relationship between BH migration and the soil physicochemical
properties (soil texture, cation exchange capacity, pH, and organic matter content), and evaluate
the influence of environmental factors (rainfall amount, solution pH, and BH solubility), humic
acid, and surfactants on BH migration in these soils. The results will provide useful data for
the rational application, safety evaluation, and risk assessment of herbicides in the rice
production system.
**2.   Materials and methods**
***2.1 Soil sampling***





In June 2020, agricultural soil samples from the surface layer (0–20 cm depth) were
collected in four major rice-producing regions throughout China, namely, Nanchang, Jiangxi
Province ($S_1$), Harbin, Heilongjiang Province ($S_2$), Ningbo, Zhejiang Province ($S_3$), and
Yichang, Hubei Province ($S_4$). For each agricultural soil, herbicide use and cultivation history
were surveyed before sampling. Based on the survey results, soil without BZB application was
collected. After collection, all soil samples were air-dried, and stones, roots, and other plant
residues were removed manually. The dry samples were ground and passed through a 2.0 mm
sieve before use. The basic physicochemical properties of each soil sample were determined
using standard soil test methods (Jackson 1958, Piper 1950, Walkley &Black 1934). The four
agricultural soils were Ferralsols ($S_1$), Phaeozems ($S_2$), Anthrosols ($S_3$), and Lixisols ($S_4$)
according to the soil classification system of the Food and Agriculture Organization of the
United Nations (FAO 1988, Nations 1998). Details of the agricultural soils are summarized in
Table 1.
***2.2 Standard solution preparation***
A sample of the BH standard (0.1250 g, accurate to 0.0001 g, 80.2% purity) was accurately
weighed into a 100-mL volumetric flask using an ATX 224 million analytical balance
(Shimadzu Corp., Kyoto, Japan). An appropriate amount of chromatographically pure
methanol was then added to dissolve the standard sample. The solution was kept in an
ultrasonic water bath using a KQ2200E Ultrasonic Cleaner (Kunshan Ultrasonic Instrument,
Kunshan, China) until clear and transparent without solids. The solution was then cooled to
room temperature and diluted with chromatographically pure methanol to prepare a 1000 mg/L
stock standard solution. Before the test, the stock standard solution was diluted with





chromatographically pure methanol to obtain a series of working standard solutions at
concentrations of 0.01, 0.05, 0.1, 0.5, 1.0, 2.5, 5.0, and 10 mg/L. Furthermore, the soil samples
were extracted with chromatographically pure acetonitrile to obtain blank soil matrix solutions.
After purification, the blank soil matrix solutions were used to dilute the BH standard solution
and give matrix standard solutions at concentrations of 0.01, 0.05, 0.1, 0.5, 1.0, 2.5, 5.0, and
10.0 mg/L. The prepared standard working solutions and matrix standard solutions were stored
in a refrigerator at 4 °C under dark conditions for later use.
**2.3.Sample preparation**
The soil sample (5.0 g) was accurately weighed into a 50-mL centrifuge tube and
moistened with ultrapure water (5.0 mL). The mixture was allowed to stand for 15 min before
adding acetonitrile (5.0 mL). After vortexing at 3000 rpm for 2 min, anhydrous $MgSO_4$ (2.0 g)
and NaCl (1.0 g) were added to the centrifuge tube. The mixture was then vortexed for 2 min
and centrifuged at 6000 rpm for 5 min. The supernatant (1.5 mL) was transferred into a 2-mL
centrifuge tube and $MgSO_4$ (150 mg) and ethylenediamine-*N*-propylsilane (PSA; 50 mg) were
added. The mixture was vortexed for 30 s and then centrifuged at 5000 rpm for 5min. Finally,
the supernatant was passed through a 0.22-μm organic filter membrane for high-performance
liquid chromatography (HPLC) analysis.
**2.4.  HPLC conditions**
Quantification of BH was performed using an Agilent 1260 high-performance liquid
chromatograph equipped with a G1329B sampler, a G1311C quaternary pump, and a G1315D
ultraviolet visible detector (Agilent Technologies, Santa Clara, CA, USA). Chromatographic
separation was achieved on a Zorbax Eclipse XDB-$C_{18}$ column (4.6 mm×150 mm, 5 μm;

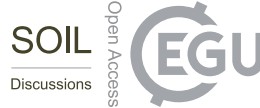

Agilent Technologies). The chromatographic conditions were as follows: Mobile phase,
methanol–0.2% phosphoric acid water (55:45, *v/v*); flow rate, 1 mL/min; detection wavelength,
286 nm; column temperature, 40 ℃; injection volume, 10 μL; and the retention time of
Glyamifop was approximately 5.45 min. Example chromatograms acquired at a spiking level
of 0.5 mg/L are shown in Fig. 1.
*2.5.  Method validation*

The mass concentrations of BH in the standard working solutions and matrix standard

solutions were measured by HPLC analysis under the indicated conditions. A standard curve
was drawn with the mass concentration of BH as the abscissa and the chromatographic peak
area as the ordinate to obtain the linear regression equation and coefficient of determination
($R^2$). The limit of detection (LOD) and limit of quantification (LOQ) of the method for BH
were determined with signal-to-noise ratios (S/N) for blank matrix extracts of 3 and 10,
respectively (Porel et al. 2014, Şengül 2016). The matrix effect (ME) was calculated by
dividing the slope of the matrix standard curve by the slope of the solvent standard curve (slope
ratio). ME > 1.1 indicates a matrix enhancement effect, ME < 0.9 indicates a matrix weakening
effect, and 0.9 < ME < 1 indicates a negligible matrix effect (Li et al. 2020). The accuracy and
precision of the BH detection method were evaluated using additive recovery and relative
standard deviation (RSD) (Ashour &Kattan 2013).
*2.6.  Herbicide mobility test in thin soil layers*

The mobility of BH in different types of agricultural soil was determined using thin-layer

chromatography (Jamet &Thoisydur 1988). The soil sample (10.0 g), which had been passed
through a 0.25-mm sieve, was accurately weighed into a 250-mL beaker and distilled water (7

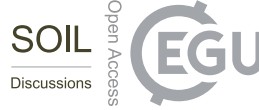

mL) was added. After stirring using a glass rod, the slurry was applied onto a glass plate (10
cm × 20 cm) and dried at 25 °C (±2 °C), with the thickness of the soil controlled between 0.5
and 1.0 mm. Subsequently, BH stock solution (10 μL, dissolved in methanol, 1000 mg/L) was
spotted at 2.5-cm intervals from the bottom of the glass plate at 25 °C (±2 °C) under light-proof
conditions. Three parallel solutions were prepared for each treatment. After the solvent had
evaporated, the thin plate was placed into a chromatography tank (20 cm × 30 cm × 30 cm)
with distilled water as the developing agent. The thin plate was unfolded to 15.0 cm and then
taken out from the tank. After drying, the soil on the thin plate was divided into six equal
segments (2.5 cm each). The BH content in the soil from each segment was measured using
HPLC analysis and its distribution on the thin plate was analyzed.
**2.7. Herbicide leaching test in soil columns**
**2.7.1. Test with different types of soil**
To explore the downward movement of BH in different types of agricultural soil, soil
column leaching tests were conducted following the OECD-312 standard method issued by the
World Economic Cooperation Organization (OECD 2004). The soil column was prepared using
a PVC pipe (inner diameter, 4 cm, length, 40 cm). A piece of filter paper was spread at the
bottom of the soil column, which was overlaid with a nylon mesh (180 μm). Next, a 1-cm-thick
quartz sand layer was added to the column, followed by the sieved soil sample (600–700 g) to
form a 30-cm deep soil column. A 0.01 mol/L $CaCl_2$ aqueous solution was added to the soil
column from the bottom to saturate the soil and remove the air through reverse osmosis. The
soil column was then hung to remove the water under gravity. The test was conducted at 25 °C
(±2 °C) and protected from light.

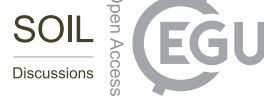

The BH stock solution was added to the top of the soil column. The dosage of BH added
to the soil column was determined according to the maximum recommended dosage of BZB
in the field and calculated using equation (1) (OECD 2004):
$$D = \frac{M \cdot 10^9 \cdot d^2 \cdot \pi}{4 \cdot 10^8}$$    (1)
where $D$ is the dosage of BH added to each soil column (μg), $M$ is the maximum recommended
dosage of BZB (kg/hm$^2$), $d$ is the soil column diameter (cm), and $\pi$ is 3.14. According to the
herbicide registration announcement, the maximum recommended dosage of BZB in rice-
growing areas is 450 g/hm$^2$. Therefore, the dosage of BH for the leaching test was calculated
to be 56 μg, and 56 μL of BH stock solution was added dropwise on the surface of each column.
After application of BH, the soil surface was covered with a 1-cm-thick quartz sand layer,
followed by a piece of filter paper and then a small amount of coarse sand. The soil column
was leached with a 0.01 mol/L CaCl$_2$ solution to simulate artificial rainfall (250 mL for 48 h)
and the leachate was collected. After leaching was complete, the soil column was evenly cut
into three sections. The BH content in the leachate and each section of the soil was measured.
*2.7.2.   Test with different herbicide dosages applied*
To analyze the influence of BH application dosage on BH leaching in the soils, three
different dosages of BH (56, 84, and 112 μg) were added to the soil columns, corresponding to
1.0, 1.5, and 2.0 times the maximum recommended dosage of BZB in rice-growing area. Briefly,
BH stock solution (56, 84, or 112 μL) was added dropwise to the surface of the soil columns.
The test procedure and sample handling were the same as described in *Section 2.7.1*. Each
treatment was repeated three times.
*2.7.3.Test with different rainfall amounts*





To investigate the depth of BH leaching in the soils after rainfall and assess the risk of BH
residues to groundwater and drinking water, the test was performed using three different
rainfall amounts (250, 500, and 1000 mL) for 48 h. The test procedure and sample handling
were the same as described in *Section 2.7.1*. Each treatment was repeated three times.
### 2.7.4.  Test with different leaching solution pH levels
To investigate the influence of different pH values on BH leaching in the soils, the test
was performed at different pH levels. Owing to acid rain caused by industrial pollution in some
areas throughout the year, rain can reach a pH as low as ~3.5. During crop production, it is
critical to improve soil fertility and increase crop yield. To increase income, fertilizers are often
applied to soil, some of which are alkaline and, therefore, increase the pH of the local soil. In
this study, pH values of 3.0, 5.0, 7.0, and 9.0 were selected as the initial leaching solution pH.
Before the test, the pH of 0.01 mol/L $CaCl_2$ solution was adjusted to 3.0±0.2, 5.0±0.2, 7±0.2,
and 9±0.2 with 0.01 mol/L HCl or NaOH. The test procedure and sample handling were the
same as described in *Section 2.7.1*. Each treatment was repeated three times.
### 2.7.5.  Test with different types of surfactant
To determine the influence of different surfactants on BH leaching in the soils, the test
was performed with leaching solutions containing critical micelle concentrations of cationic
surfactant (cetyl trimethyl ammonium bromide, CTAB), anionic surfactant (dodecyl benzene
sulfonic acid, SDBS), and nonionic surfactant (Tween-80). The soil column leached with a
surfactant-free solution was used as a control. The test procedure and sample treatment were
the same as described in *Section 2.7.1*. Each treatment was repeated three times.
### 2.7.6.  Test with different humic acid concentrations



243 To investigate the influence of humic acid on BH leaching in the soils, the concentration

244 of humic acid (≥90%, Macklin Biochemical Co., Ltd., Shanghai, China) was set to 0.5%, 1%,

245 and 2% under the range of organic matter content in the experimental soils (0.23–2.04%; Table

246 1). The soil column without humic acid added was used as a control. The test procedure and

247 sample handling were the same as in *Section 2.7.1*. Each treatment was repeated three times.

248 **2.8.  Data analysis**

249 The mobility retention factor ($R_f$) of BH on the thin plate was calculated using equation (2):

$$R_f = \frac{\sum Z_i \times M_i}{Z_w \times \sum M_i} \qquad (2)$$

251 where $i$ is the number of segments in which the plate is divided, $Z_i$ is the distance of BH in

252 segment $i$ from the origin, $Z_w$ is the distance of the solvent front from the origin, and $M_i$ is the

253 soil BH content in segment $i$ (Haskis et al. 2019).

254 The BH contents in each soil section and the leachate were calculated as the percentage

255 of BH recovered from the total amount of BH added using equation (3):

$$R_i = \frac{m_i}{m_o} \times 100 \qquad (3)$$

257 where $R_i$ is the mass fraction of BH in section $i$ of the soil or leachate in the total mass of BH

258 added; $m_i$ is the mass of BH in each section of soil and leachate (mg); $i$ = 1, 2, 3, and 4, which

259 represents the 0–10 cm, 10–20 cm, and 20–30 cm soil sections, and the leachate, respectively;

260 $m_0$ is the total mass of BH added (mg). According to the value of $R_i$, the mobility of BH in the

261 soil was divided into four classes: (I) $R_4$ > 50%, easy to leach; (II) $R_3 + R_4$ > 50%, moderate

262 leaching; (III) $R_2 + R_3 + R_4$ > 50%, difficult to leach; and (IV) $R_1$ > 50%, very difficult to leach.

263 Data processing and statistical analysis were performed in SPSS Statistics 22.0 (IBM SPSS,

264 Armonk, NY, USA). The adsorption curves of Glyamifop were plotted using OriginPro 8.0



(OriginLab Corp., Northampton, MA, USA).

## 3.    Results and discussion

### 3.1. Linearity, sensitivity, matrix effect, accuracy, and precision of the method

In different matrices (methanol, water, and soils), a good linear relationship was observed
between the peak area and the mass concentration of BH within the range of 0.01–10 mg/L
($R^2 >0.99$; Table 2). The LOD and LOQ for BH in water and soils were in the ranges of 6.3–
16.5 μg/kg and 20.8–55 μg/kg, respectively. The ME of BH in the four different types of
agricultural soil was greater than 1.1, indicating that the matrix effect can be ignored. The
average recovery of BH in each soil matrix was between 81.79% and 101.09%, with an RSD
of 0.56%–7.87%. The accuracy and precision of the established method met the requirements
of herbicide residue analysis (70% < recovery < 110%, RSD < 20%) (NY 2014).

### 3.2. Migration characteristics of BH in thin soil layers

The $R_f$ values of BH in $S_1$ to $S_4$ were 0.45, 0.34, 0.75, and 0.90, respectively (Table 3),
indicating that the mobility of BH in the four types of agricultural soil was in the order $S_4 >$
$S_3 > S_1 > S_2$. According to the grading standard in the "Test Guidelines on Environmental Safety
Assessment for Chemical Pesticides—Soil Leaching Test" [45], BH is moderately mobile in
Ferralsols and less mobile in Phaeozems, while it is mobile in Anthrosols and extremely mobile
in Lixisols.
Soil physicochemical properties are important factors influencing the migration behavior
of herbicides in soil (Liu et al. 2018). Linear correlation analysis revealed the relationships
between the $R_f$ value of BH and the physicochemical properties of agricultural soils (Table 4).
The $R_f$ value of BH was positively correlated with both soil sand content and pH (slope > 0).



The correlation with soil sand content was significant ($P = 0.027$), while that with soil pH was
not significant ($P = 0.285$). In contrast, the $R_f$ value of BH was negatively correlated with soil
silt content, clay content, cation exchange capacity (CEC), organic carbon (OC) content, and
organic matter (OM) content (slope < 0), but these correlations were not significant ($P = 0.681$,
0.152, 0.258, 0.181, and 0.160, respectively). Previous research has indicated that the BH
adsorption capacity of agricultural soils is positively correlated with soil clay content, CEC,
OC content, and OM content (Rao et al. 2020). Therefore, our results confirmed that the
migration capacity of BH in agricultural soils is negatively correlated with the soil adsorption
capacity of BH.
***3.3. Leaching characteristics of BH in soil columns***
***3.3.1. Influence of soil type on BH leaching and migration***

The leaching of herbicides in the soil is an important process determining whether

herbicides enter groundwater and cause pollution (Younes &Galal-Gorchev 2000). The soil
column leaching test results showed that the leaching and migration characteristics of BH
varied considerably with different soil types (Fig. 2.). After leaching with $CaCl_2$ solution (250
mL), BH in the $S_1$ soil columns was mainly distributed in the 0–20 cm sections, with the
maximum BH content in the 10–20 section. In the $S_2$ soil columns, BH was also mainly
distributed in the 0–20 cm sections, but the BH content in these sections was markedly higher
than that of $S_1$, while the maximum BH content occurred in the 0–10 cm section. For soils $S_3$
and $S_4$, BH was mainly distributed in the leachate, with the leaching rate of BH in $S_4$ greater
than that in $S_3$.

Based on the $R_i$ values, the leaching rate of BH in the four agricultural soils was in the



order $S_4 > S_3 > S_1 > S_2$ (Table 5). According to the "Test Guidelines on Environmental Safety
Assessment for Chemical Pesticide" (NY 2014), BH was relatively difficult to leach in
Ferralsols, difficult to leach in Phaeozems, and easy to leach in Anthrosols and Lixisols. The
leaching and migration characteristics of BH in the four agricultural soils were distinctly
different, indicating the strong influence of soil physicochemical properties on the
environmental behavior of this herbicide. For example, BH not being detected in the leachate
of $S_2$ might be attributed to the high organic matter content (Table 1) and strong BH adsorption
capacity of the Phaeozems, which blocked BH migration.
*3.3.2. Influence of herbicide dosage on BH leaching and migration*

The speed and depth of the downward migration of herbicides in soil are closely related

to their adsorption process in the soil (Oliveira Jr et al. 2001). Therefore, the factors influencing
the adsorption process will also influence the herbicide migration process. In addition to soil
physicochemical properties, the application dosage of herbicides is an important factor
influencing their leaching and migration in soil (Ferrero et al. 2001). After applying BH at
different dosages, the distribution and content of this herbicide in the soil sections and leachate
changed considerably (Fig. 3.). When 56 µg of BH was applied, BH in the $S_1$ soil columns was
mainly distributed in the 0–20 cm sections and the maximum BH content was located in the 0–
10 cm section. When the dosage of BH was increased to 84 and 112 µg, BH was still mainly
distributed in the 0–20 cm sections of $S_1$, but the maximum BH content moved down to the
10–20 cm section. In particular, after applying BH at the high dosage of 112 µg, BH seeped
out of the soil column and was detected in the leachate. In the $S_2$ soil columns that received 56
µg of BH, all BH was distributed in the 0–20 cm sections after leaching. However, when the





BH dosage was increased to 84 and 112 μg, the herbicide migrated down to the 20–30 cm
section, but none was detected in the leachate. In the $S_3$ and $S_4$ soil columns, the maximum BH
content was observed in the leachate. Furthermore, with increasing BH dosage, the BH content
in each soil section decreased, while the BH content in the leachate increased.

When the application dosage of BH increased from 56 to 112 μg, the BH content in the 0–

10 cm section of the four agricultural soils decreased and migrated to lower sections of the soil
column. Except for $S_2$, the BH content in the leachate of different soils increased, indicating
that the application dosage of the herbicide influenced its migration behavior in the soil. A
higher concentration of the herbicide resulted in greater migration depth into the soil and higher
leachability. A plausible explanation is that the soil has a fixed number of effective adsorption
sites for the herbicide. When the BH dosage increases, the soil adsorption sites are occupied
and cannot adsorb additional BH molecules, leading to downward migration along with the
leachate.
### 3.3.3. Influence of rainfall amount on BH leaching and migration

The migration behavior of herbicides in soil mainly includes upward, downward, and

lateral migration, of which downward migration, driven by the effect of gravity on seepage
water, is dominant (Oppong &Sagar 1992). Therefore, rainfall can influence the migration
behavior of herbicides in soil. The results of the leaching test under different rainfall conditions
showed that BH leaching and migration in the four types of agricultural soil were influenced
by the rainfall amount (Fig. 4). When the simulated rainfall was 250 mL, BH was mainly
distributed in the soil sections of $S_1$, with the maximum BH content in the 10–20 cm section.
In $S_2$, all BH was distributed in the soil sections, and the maximum BH content was located in





the 0–10 cm section. Limited BH was distributed in the soil sections of $S_3$ and $S_4$, while the
majority of the herbicide was detected in the leachate of these two soils.

When the simulated rainfall amount was increased to 500 mL, the BH content in the

leachate increased for all different soils, and the depth of the maximum BH content in $S_1$ and
$S_2$ moved downward by 10 cm. When the simulated rainfall was further increased to 1000 mL,
the depth of the maximum BH content in $S_1$ and $S_2$ changed to the 20–30 cm section. However,
no BH was detected in the soil sections of $S_3$ and $S_4$, and all BH was leached from the soil and
distributed in the leachate (Fig. 4.). The rainfall intensity was positively correlated with the
depth and amount of BH leaching in the soil. This result indicated that with an increasing
amount of rainfall, more BH migrated to a greater depth in the soil, and the leaching rate of BH
from the soil also increased.
*3.3.4. Influence of solution pH on BH leaching and migration*

The pH can influence the activity of soil microorganisms, the conversion of soil organic

matter, and the existence of soil substances. Furthermore, pH affects the physicochemical
properties of herbicides, directly or indirectly influencing their adsorption, migration,
transformation, and enrichment processes in soil (Walker &WELCH 1989). Triketone
herbicides such as BH are weak acids (pKa ~3) that are stable under acidic conditions, but
relatively unstable in neutral and alkaline media (Williams &Tjeerdema 2016). The leaching
test results showed that changes in the pH value of the leaching solution strongly influenced
the downward mobility of BH in the soil (Fig. 5.). Under different pH conditions, the amount
of BH migrating from the soil was largest in $S_4$. When the leaching solution pH was 9, the
amount of BH in the leachate in the four soils was markedly higher than that in the





corresponding soils at pH 3, 5, and 7. This result indicated that the migration ability of BH was
stronger in alkaline environments compared with neutral and acidic environments.

When the leaching solution pH was 3 and 5, BH in the soil columns of $S_1$ and $S_2$ was

mainly distributed in the 0–20 cm sections, and the maximum BH content was located in the
0–10 cm section. In contrast, the maximum BH content in the soil columns of $S_3$ and $S_4$ was
located in the 20–30 cm section, and the BH content of the leachate was greater than that of
any soil section. When the leaching solution pH was 7, the depth of maximum BH content in
the soil columns did not change, except for the soil column of $S_1$, in which it moved down to
the 10–20 cm section. In summary, changing the solution pH significantly influenced the
mobility of BH in the soil columns, with a higher solution pH resulting in easier BH migration
downward, which was more likely to pose a threat to groundwater quality.
*3.3.5. Influence of surfactant type on BH leaching and migration*

The interaction of surfactants and herbicides is a complicated process. In water–soil

systems, surfactants can change the soil physicochemical properties, such as the surface tension
of soil water, capillary diffusion, water holding capacity, osmosis, pH value, and CEC,
influencing the environmental behavior of herbicides in soil (Haigh 1996). In the present study,
the leaching test results showed that various types of surfactant had distinctly different
influences on BH leaching and migration in the four agricultural soils (Fig. 6.). Compared with
the control without surfactant treatment, the BH content was much higher in the 0–10 cm
section of CTAB-treated soil columns, while the BH content in the leachate of CTAB-treated
soil columns was lower. These results indicated that cationic surfactant CTAB had a delaying
effect on BH leaching and migration in the soil. This might be due to the soil itself being



negatively charged, such that the cationic surfactant is replaced by inorganic cations in the soil
and then adsorbed by the soil, enhancing the herbicide adsorption capacity of the soil (Sheng
et al. 1996, Zhu et al. 2000). Therefore, BH was adsorbed and fixed in the soil under the action
of the cationic surfactant, blocking BH migration in the soil.
When anionic surfactant SDBS or nonionic surfactant Tween-80 was added to the leaching
solution, the amount of BH leached from the four agricultural soils was higher than that in the
control. The leached amount of BH in SDBS-treated soil columns was greater than that in
Tween-80-treated soil columns (Fig. 6.). Specifically, in the control soil columns of $S_1$ and $S_2$,
BH was mainly distributed in the 0–20 cm sections, and the maximum BH content was located
in the 10–20 cm and 0–10 cm sections, respectively. When SDBS or Tween-80 was added to
the leaching solution, BH in the soil columns of $S_1$ was mainly distributed in the 10–30 cm
sections, with the maximum BH content located in the 20–30 cm section. The BH content in
the 0–10 cm section of $S_2$ was lower than that in the control. The BH content in the soil columns
of $S_3$ and $S_4$ was mainly distributed in the leachate, and the BH content in each soil section was
lower than that in the control. These results indicated that adding anionic surfactant SDBS and
nonionic surfactant Tween-80 was an effective approach to promoting BH leaching and
migration in the soil, with SDBS having a stronger effect than Tween-80. In general, herbicides
with higher water solubility show stronger leaching and migration abilities in soil. Anionic and
nonionic surfactants can produce a large number of micelles in water, which increases the
amount of OM and herbicide in the soil. Soil OM is dissolved in the water phase and the OM
content in the soil phase decreases, which weakens the herbicide adsorption ability of the soil.
However, anionic and nonionic surfactants increase the water solubility of the herbicide,



facilitating its migration in the soil. Therefore, both anionic and nonionic surfactants increase
the solubility of BH in water and weaken BH adsorption by the soil, resulting in enhanced
herbicide leaching and migration in the soil.

### 3.3.6. Influence of humic acid concentration on BH leaching and migration

Humic acid is a natural weakly acidic macromolecular organic acid formed by the
decomposition and transformation of animal and plant residues. Humic acid has a network
structure and is the main adsorption center of OM in soil. In general, the adsorption capacity
of soil is directly proportional to its OM content, and influences the environmental behavior of
organic compounds, such as herbicides in soil (Álvarez-Benedí et al. 1998). In this study, the
results showed that adding different concentrations of humic acid strongly influenced BH
leaching and migration in the agricultural soils (Fig. 7.). In the control without humic acid
addition, BH in the soil columns of $S_1$ and $S_2$ was mainly distributed in the 0–20 cm sections,
with the maximum BH content located in the 10–20 cm section of $S_1$ and the 0–10 cm section
of $S_2$. When 0.5% humid acid was added, the distribution and content of BH in the soil columns
of $S_1$ and $S_2$ changed, and the BH content in each soil section was higher than that of the control.
When the concentration of humic acid added was increased to 1% and 2%, the BH content in
the 0–10 cm sections of both $S_1$ and $S_2$ increased, and the depth of the maximum BH content
in $S_1$ changed to 0–10 cm section.
Regarding $S_3$ and $S_4$, BH in the soil columns of the control was mainly distributed in the
leachate. As the concentration of humic acid added to the soil increased, the BH content in the
leachate gradually decreased and the BH content in each section of the soil columns was higher
than that in the control (Fig. 7.). These results indicated that adding humic acid to the soil





blocked downward leaching and migration of BH in the agricultural soils, with this blocking
effect increasing with an increasing concentration of humic acid. We speculated that adding
humic acid to the soil reduced the soil pH and increased the soil OM content. According to
previous research, the ability of soil to adsorb BH is negatively correlated with soil pH and
positively correlated with soil OM content. Therefore, adding humic acid enhanced the BH
adsorption capacity of the soil, blocking BH leaching and migration into the deeper soil.
**4.    Conclusions**
This study explored the leaching and migration behavior of benzobicyclon hydrolysate
in four different types of agricultural soil in China using soil thin-layer chromatography and
column leaching tests. The results indicated that this herbicide was moderately or less mobile
and difficult to leach in Ferralsols and Phaeozems, and highly mobile and easily leached in
Anthrosols and Lixisols. Among the four types of agricultural soil, benzobicyclon hydrolysate
had the lowest migration ability in Phaeozems and appeared safe to surface water and
underground water. However, it possessed a moderate migration ability in Ferralsols, posing
a threat to groundwater and drinking water. In Anthrosols and Lixisols, this herbicide showed
a high migration ability and might pollute groundwater and drinking water.
**Ethical Approval (Not applicable)**
**Consent to Participate**
All of the authors consent to participate in drafting of this manuscript
**Consent to Publish**
All of the authors consent to Publish this manuscript
**Authors Contributions**
Lang Liu: Conceptualization, Methodology, Software, Investigation, Writing -Original
Draft
Lei Rao: Validation, Formal analysis, Visualization
Wenwen Zhou: Writing -Review & Editing



Limei Tang:Writing -Review & Editing
Baotong Li: Resources, Writing -Review & Editing, Supervision Data Curation

**Funding**
This study was supported by the "13th Five-Year" National Key Research Program of
China (Grant No. 2017YFD0301604).
**Competing Interests**
The authors declare no competing financial interest.
**Availability of data and materials**
All data generated or analysed during this study are included in this published article

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



**Table 1**
Sampling sites and basic physicochemical properties of four different agricultural soils in
China

| Soil | Site (latitude, longitude) | Classification | Texture | | | Type | pH | CEC (cmol/kg) | OC (%) | OM (%) |
|------|---------------------------|----------------|---------|---------|---------|------|------|------|------|------|
| | | | Sand (%) | Silt (%) | Clay (%) | | | | | |
| $S_1$ | Nanchang, Jiangxi (N28°46', E115°36') | Ferralsols | 26.68 | 14.39 | 58.93 | Clay | 4.74 | 12.50 | 0.40 | 0.69 |
| $S_2$ | Haerbin, Heilongjiang (N41°36', E127°53') | Phaeozems | 18.52 | 36.24 | 45.24 | Sandy loam | 6.24 | 34.18 | 1.35 | 2.04 |
| $S_3$ | Ningbo, Zhejiang (N29°14', E121°48') | Anthropos | 41.16 | 22.24 | 36.60 | Loam | 7.06 | 14.98 | 0.24 | 0.42 |
| $S_4$ | Yichang, Hubei (N35°06', E118°21') | Lixisols | 62.75 | 23.18 | 14.07 | Sandy | 7.17 | 9.34 | 0.14 | 0.23 |

CEC, cation exchange capacity; OC, organic carbon content; and OM, organic matter content.
















**Table 2**
Linear equation, coefficient of determination ($R^2$), limit of detection (LOD), limit of
quantification (LOQ), matrix effect (ME), accuracy and precision of benzobicyclon
hydrolysate in different matrices

| Matrix | Regression equation | $R^2$ | LOD (mg/kg) | LOQ (mg/kg) | ME | Average recovery (%) | | | Relative standard deviation (%) | | |
|---|---|---|---|---|---|---|---|---|---|---|---|
| | | | | | | 0.1 mg/kg | 0.5 mg/kg | 1.0 mg/kg | 0.1 mg/kg | 0.5 mg/kg | 1.0 mg/kg |
| Methanol | y=23.419x+0.916 | 1.0000 | - | - | - | - | - | - | - | - | - |
| Water | y=21.886x+1.145 | 0.9992 | 0.0063 | 0.0208 | 0.93 | 101.09 | 97.75 | 97.61 | 6.18 | 1.99 | 3.62 |
| $S_1$ | y=23.855x-0.493 | 0.9999 | 0.0139 | 0.0464 | 1.02 | 88.19 | 87.21 | 84.16 | 3.38 | 1.41 | 2.52 |
| $S_2$ | y=23.452x+0.699 | 0.9999 | 0.0150 | 0.0501 | 1.00 | 87.68 | 81.79 | 84.16 | 4.67 | 4.38 | 1.92 |
| $S_3$ | y=25.785x-0.643 | 0.9998 | 0.0067 | 0.0223 | 1.10 | 94.48 | 93.72 | 94.82 | 7.87 | 3.15 | 0.56 |
| $S_4$ | y=23.567x+3.330 | 0.9990 | 0.0165 | 0.0550 | 1.01 | 95.49 | 92.63 | 92.02 | 3.38 | 4.11 | 0.76 |






**Table 3**
Distribution and content of benzobicyclon hydrolysate in thin soil layers

| Soil | Benzobicyclon hydrolysate content (μg) | | | | | | $R_f$ |
|------|--------|--------|--------|--------|--------|--------|------|
| | 0–2.5 cm | 2.5–5 cm | 5–7.5 cm | 7.5–10 cm | 10–12.5 cm | 12.5–15 cm | |
| $S_1$ | 1.67±0.03 | 2.36±0.08 | 3.2±0.13 | 1.44±0.04 | 0.59±0.02 | 0.17±0.01 | 0.45 |
| $S_2$ | 2.99±0.27 | 4.3±0.31 | 1.4±0.08 | 0.89±0.05 | 0.11±0.01 | ND | 0.34 |
| $S_3$ | 0.07±0.01 | 0.3±0.02 | 1.09±0.09 | 2.48± | 3.86±0.31 | 1.44±0.11 | 0.75 |
| $S_4$ | 0.05±0.01 | 0.18±0.01 | 0.23±0.01 | 0.64±0.03 | 2.71±0.21 | 5.67±0.46 | 0.90 |

$S_1$ to $S_4$ are defined in Table 1. $R_f$, mobility retention factor. Values are the means (n = 3).



**Table 4**

Linear correlations between the mobility retention factor ($R_f$) of benzobicyclon hydrolysate and

the physicochemical properties of agricultural soils

| Soil property | Sand (%) | Silt (%) | Clay (%) | pH | CEC (cmol/kg) | OC (%) | OM (%) |
|---|---|---|---|---|---|---|---|
| Slope | 72.654 | −11.135 | −61.520 | 3.089 | −31.765 | −1.752 | −2.649 |
| Intercept | −7.042 | 30.708 | 76.237 | 4.418 | 37.126 | 1.601 | 2.461 |
| significance level | 0.027 | 0.681 | 0.152 | 0.285 | 0.263 | 0.181 | 0.160 |
| Correlation coefficient | 0.919 | 0.102 | 0.720 | 0.511 | 0.543 | 0.671 | 0.706 |

$S_1$ to $S_4$ are defined in Table 1. $R_f$, The mobility retention factor; CEC, cation exchange capacity;

OC, organic carbon content; and OM, organic matter content

**Table 5**

The content ($R_i$) of benzobicyclon hydrolysate (BH) in the soil and leachate samples after

leaching with 250 mL of CaCl$_2$ solution for 48 h

| Soil | $R_1$ | $R_2$ | $R_3$ | $R_4$ | $R_3+R_4$ | $R_2+R_3+R_4$ |
|---|---|---|---|---|---|---|
| $S_1$ | 26.87±1.01 | 43.48± 1.58 | 20.91±0.89 | 8.74 ±0.24 | 29.65 | 73.13 |
| $S_2$ | 66.35± 2.35 | 25.63±0.92 | 8.02±0.46 | 0.00 ±0.00 | 8.02 | 33.65 |
| $S_3$ | 4.54±0.12 | 10.45±0.54 | 21.49±1.01 | 63.52±1.93 | 85.01 | 95.46 |
| $S_4$ | 2.11±0.08 | 4.11±0.36 | 11.71±0.69 | 82.06±2.56 | 93.78 | 97.89 |

$S_1$ to $S_4$ are defined in Table 1. $R_1$, $R_2$, $R_3$, and $R_4$ represent the mass fraction of BH recovered

from the 0–10, 10–20, and 20–30 cm soil sections and the leachate in the total amount of BH

added, respectively. Values are the mean (n = 3).



**Figure captions**
**Fig. 1.** Typical chromatograms of benzobicyclon hydrolysate (BH) in methanol and soil. A.
Blank sample of $S_4$; B. 0.5 mg/L standard product of BH; C. Soil sample spiked with 0.5 mg/L
BH
**Fig. 2.** The distribution and content of benzobicyclon hydrolysate (BH) in soil columns of four
different soil types after leaching with 250 mL of $CaCl_2$ solution for 48 h ($S_1$ to $S_4$ are defined
in Table 1). Ri represents the mass fraction of BH recovered from the 0–10, 10–20, and 20–30
cm soil sections or the leachate in the total amount of BH added, respectively. Values are the
means ± standard error (n = 3).
**Fig. 3**. The distribution and content of benzobicyclon hydrolysate in soil columns after leaching
under different application dosages of benzobicyclon hydrolysate ($S_1$ to $S_4$ are defined in Table
1). Values are the means ± standard error (n = 3).
**Fig. 4.** The distribution and content of BH in soil columns after leaching with different rainfall
amounts ($S_1$ to $S_4$ are defined in Table 1). Values are the means ± standard error (n = 3).
**Fig. 5.** The distribution and content of BH in soil columns after leaching with different pH
levels ($S_1$ to $S_4$ are defined in Table 1). Values are the means ± standard error (n = 3).
**Fig. 6.** The distribution and content of BH in soil columns after leaching in the presence of
different surfactants ($S_1$ to $S_4$ are defined in Table 1). Values are the means ± standard error (n
= 3).
**Fig. 7.** The distribution and content of BH in soil columns after leaching with the addition of
different concentrations of humic acid ($S_1$ to $S_4$ are defined in Table 1). Values are the means ±
standard error (n = 3).



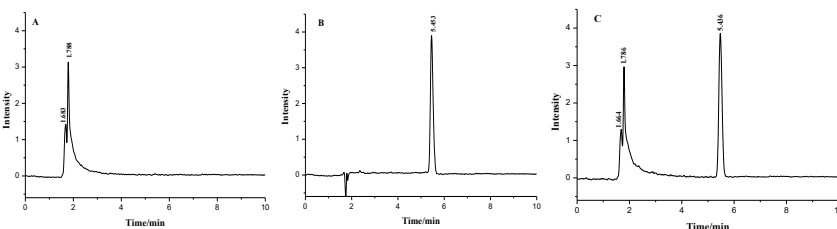


**Fig. 1.** Typical chromatograms of benzobicyclon hydrolysate (BH) in methanol and soil. A.

Blank sample of $S_4$; B. 0.5 mg/L standard product of BH; C. Soil sample spiked with 0.5 mg/L

BH

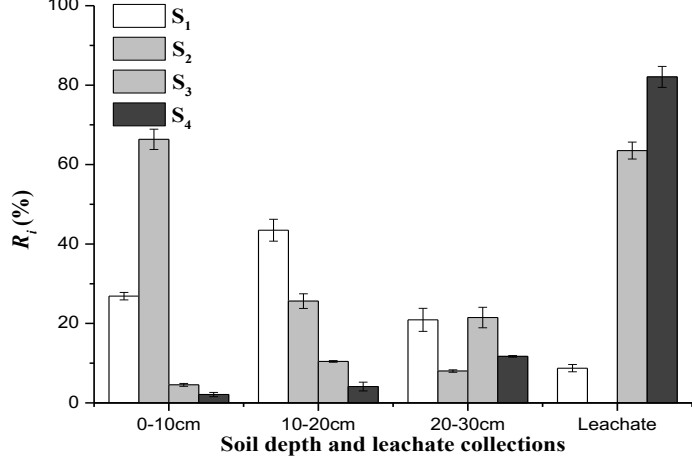

706

**Fig. 2.** The distribution and content of benzobicyclon hydrolysate (BH) in soil columns of four

different soil types after leaching with 250 mL of $CaCl_2$ solution for 48 h ($S_1$ to $S_4$ are defined

in Table 1). Ri represents the mass fraction of BH recovered from the 0–10, 10–20, and 20–30

cm soil sections or the leachate in the total amount of BH added, respectively. Values are the

means ± standard error (n = 3).

712





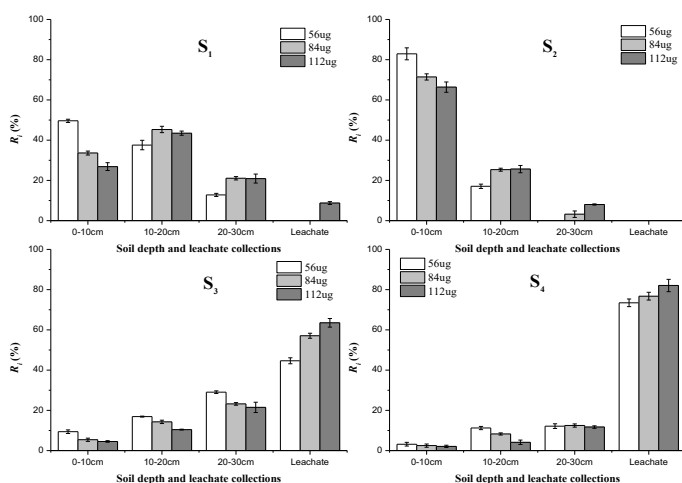

**Fig. 3**. The distribution and content of benzobicyclon hydrolysate in soil columns after leaching

under different application dosages of benzobicyclon hydrolysate ($S_1$ to $S_4$ are defined in Table

1). Values are the means ± standard error (n = 3).

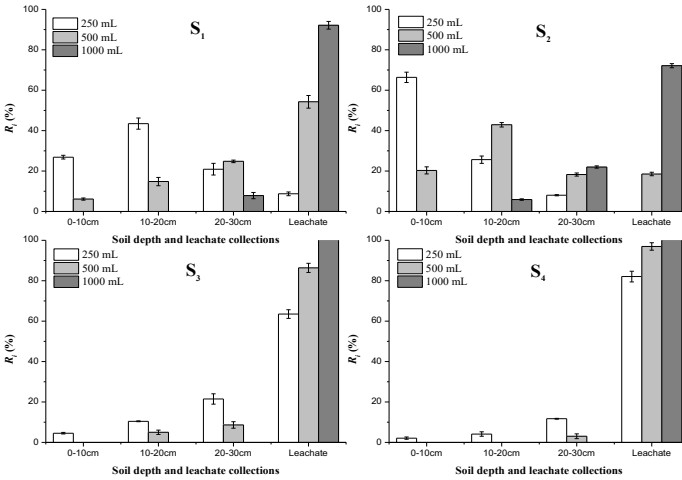

**Fig. 4.** The distribution and content of BH in soil columns after leaching with different rainfall

amounts ($S_1$ to $S_4$ are defined in Table 1). Values are the means ± standard error (n = 3).



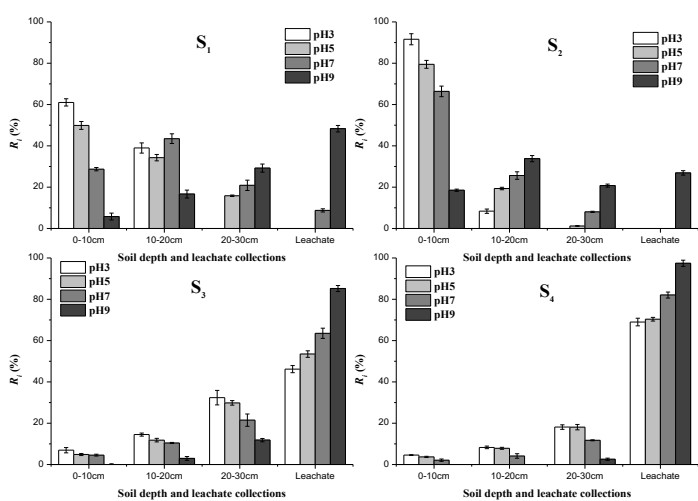

**Fig. 5.** The distribution and content of BH in soil columns after leaching with different pH

levels ($S_1$ to $S_4$ are defined in Table 1). Values are the means ± standard error (n = 3).

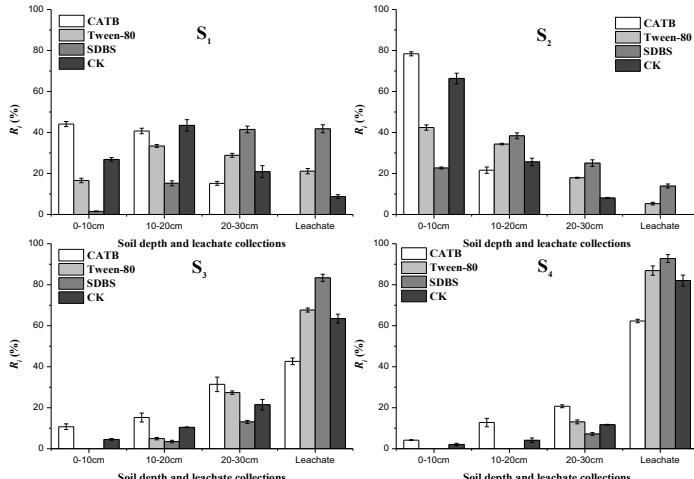

**Fig. 6.** The distribution and content of BH in soil columns after leaching in the presence of

different surfactants ($S_1$ to $S_4$ are defined in Table 1). Values are the means ± standard error (n

= 3).





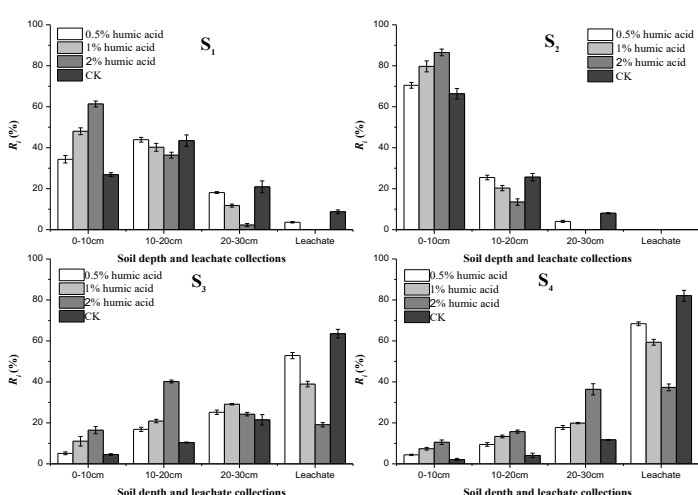

729

**Fig. 7.** The distribution and content of BH in soil columns after leaching with the addition of

different concentrations of humic acid ($S_1$ to $S_4$ are defined in Table 1). Values are the means ±

standard error (n = 3).

733

734