# Peer review of "Migration behavior of benzobicyclon hydrolysate and associated influencing"

_SOIL, 2021_

## Author Comment (AC1)

**Soil**

**_Comment:_** "Several grammatical and syntax issues are present within the manuscript, as well as many incorrect statements (e.g. mentioning non-significant results in the abstract and saying an increase in BH dosage and rainfall amount blocked BH migration in the abstract). These issues have not been outlined, however, in light of two major issues I have with the project design which makes me sceptical of the results and if cannot be fixed is reason to reject the manuscript.

1) The main conclusion of the manuscript is the ranking of four soil types in regards to the mobility of BH, and their susceptibility to leaching BH. However, only a single sample has been taken for each soil type and as such, there is no measure of variability within the soil types that would be necessary to make sweeping statements about the soil type as a population. Due to this, the conclusions like:

"Based on the mobility retention factor (Rf = 0.34–0.90), the mobility of BH in thin soil layers was ranked in the order Lixisols > Anthrosols > Ferralsols > Phaeozems"

would need to be changed to:

"Based on the mobility retention factor (Rf = 0.34–0.90), the mobility of BH in thin soil layers was ranked in the order **_S4 > S3 > S1 > S2_**"

as you can only justifiably say that there is a difference between the samples as opposed to soil types.

 **_Response:_**  **_Thank you for your comments. Some of the statements in the summary do have errors. We have revised the statements in the summary and revised them as follows according to your requirements:_**Based on the mobility retention factor ($R_f$ = 0.34–0.90), the mobility of BH in thin soil layers was ranked in the order S4 > S3 > S1 > S2. The $R_f$ value of BH was linearly positively correlated with soil sand content and pH, and negatively correlated with other physical and chemical properties of soil. BH was difficult to leach in S2, less difficult to leach in S1, and easy to leach in S3 and S4. Increasing the BH dosage, rainfall amount, leaching solution pH and adding anionic (dodecyl benzene sulfonic acid) or nonionic (Tween-80) surfactant promoted BH migration in soil columns. In contrast, increasing the adding humic acid and cationic surfactant (cetyl trimethyl ammonium bromide) blocked BH migration in soil columns. **_And modify the soil type to S1-S4 in the full text_**

**_Comment:_**    The method indicates that 4 cm diameter by 30 cm cores were packed with 600 to 700 g of air-dried soil. A core packed with 600 g of soil would have an air-dried bulk density of 1.59 g/cm^3, while a core packed with 700 g would have an air-dried bulk density of 1.86 g/cm^3. Thus, cores have a potential variation of 15-17% in air-dried bulk

density which would have significant impacts on the porosity of the cores, the dynamics of water and thus the dynamics of the solute. This issue in the method would make me sceptical of any of the results from the leaching experiments.

-Additional questions about the method were if the soil was uniformly dried? Was the water content of the soil at packing determined? Because variation in moisture content at packing can introduce significant artefacts especially when dealing with variable soils.

 *Response:*

*All the soil was spread on the floor before the column was filled with ventilation and dried, then ground and sieved, and finally the moisture content of each soil after drying was measured by the drying method, and the moisture content of S1, S2, S3, and S4 were 3.5 %, 4.7%, 1.7%, 2.1%. All water contents are below 5%, and the impact on the soil column leaching test is almost negligible; thank you for your opinion, we have added this data to Table 1; another On the other hand, because the mechanical composition of different soils is different, and the content of sand and clay particles in different soils is different, the bulk density of the main soil after filling the column is different, resulting in the same volume of soil quality will be inconsistent, appearing 600-700g The quality within the range. However, all the soil in the soil pillar is the soil after being compacted, and it will not artificially cause the gap to be not filled.*

---

## Author Comment (AC2)

**Soil**

grammar and writing style: the whole paper is affected by several grammatical issues with improper use of the scientific English style (see the syntax);
response: We have hired a professional to polish the language of the article

Introduction: too long and not really focused on paper background;
response: We expounded the leaching hazards of herbicides in soil, as well as some specific factors (background) affecting leaching, so we need to study the specific leaching behavior of the herbicides (purpose), and finally introduce the research methods (means). It is considered to be concerned with the background of the article, please tell us the detail

- Materials and Methods: too questionable approaches. For example, authors collected soil samples from the Ap surface horizon (I suppose: 0–20 cm depth; NOT LAYER!) from four major rice-producing regions throughout China. However, as clearly shown by soil classification, this method introduced a significant variability that the authors do not take into account or justify in their results. Additionally, from a statistical analysis viewpoint, the authors just stated that "statistical analysis were performed in SPSS Statistics 22.0", without explaining which kind of data treatment they used. This is a dramatic issue since without explaining statistical approach and previous data pretreatment, this Reviewer, as every other serious Reviewer worldwide, cannot judge if obtained outcomes are affected by mistakes in data treatments;

response: we have change to the surface horizon; and we used the LSD for standard deviation tests, and Linear Regression for linear correlations between different factors and leaching

- Results and Discussion: results and the following discussion are too speculative and very often justified by using very old references or without any comparison with previous studies. In this last case, this is not due to the novelty of the paper's outcomes but just, in this Reviewer's opinion, to the overspeculative comments of the authors.

response: Before this article, we studied the adsorption of BH in soil and the influencing factors(Adsorption–desorption behavior of benzobicyclon hydrolysate in different

agricultural soils in China, Ecotoxicology and Environmental Safety 202 (2020) 110915). This article wants to verify the relationship between the adsorption results and the leaching results, so this article is studied, so this article is based on the discussion in the previous paper. description, so I don't think there are speculative comments